# The Yonder Man and the Hypocrite in Seneca's *Epistle* 59 and Paul's Letter to the Romans

Joseph R. Dodson 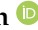

Denver Seminary, Littleton, CO 80120, USA; joey.dodson@denverseminary.edu

**Abstract:** Scholars have long recognized how Romans 1–2 is replete with resonances of Stoic traditions as they have referred to specific similarities in Seneca's writings and the impact on the interpretation of the letter. Nevertheless, a significant parallel to Paul's polemic against his fictitious opponent in Rom 2:17–24 has been neglected, namely, Seneca's invective in *Epistle* 59. There, the Stoic calls out the "yonder man," who harms others despite being known as "most gentle"; who robs others despite being considered "most generous"; and who engages in drunkenness and lust despite his reputation of being "most-temperate." This parallel is also relevant because, like that of Romans 2, the larger context of *Epistle* 59 also regards human depravity. Therefore, in this article, I will seek to buttress the conclusions from scholars regarding how well Romans 2 aligns with passages from Seneca. I will also aim to show, however, that—in contrast to Paul—Seneca shows solidarity with his interlocutor by recognizing his own shortcomings. Hence, while the similarities help scholars understand how Stoic traditions impact the creation and interpretation of Romans, the convergence between *Epistle* 59 and Romans 2 also highlights their great divergence. Thus, while the comments in *Epistle* 59 support the arguments regarding Stoic influence in Romans, the parallels remind the scholar that even as Paul draws upon Stoic ideas and rhetorical devices to deride his interlocutor, he would also consider himself and his fellow believers as not only distant from the likes of the pretentious yonder man but from the humble hypocrisy of Seneca too.

**Keywords:** Romans 2; *Epistle* 59; Seneca; diatribe; interlocutor

## 1. Introduction

Scholars have long considered Paul's rhetoric in Romans 1–2 as consisting of "Jewish-tinged Stoic polemics" (*jüdisch gefärbter stoischer Polemik*) which show how Stoic traditions color the creation and interpretation of the letter (Fridrichsen 1927, p. 40). For instance, in his foundational work, *Der Stil des Paulus und die Diatribe*, Rudolf Bultmann discusses how Paul's use of rhetorical devices in Romans parallels the Stoic address to the "foolish student" and "fictitious opponent" in "didactic debates" (Bultmann 1910, p. 66). Following his professor, Ernst Käsemann also concludes that Romans 1–2 contains "*erstaunliche*" and "*ausgezeichneten Parallelen*" with Stoicism and refers to Seneca's *Epistle* 39 as a case in point (Käsemann 1974, pp. 48, 65). There, the sage writes about how people sink themselves in pleasures until they cannot do without them—to the degree that what once was superfluous for them has become indispensable to them. People are "most wretched," then, according to Seneca, because rather than enjoying their pleasures, they are enslaved to their passions. Humanity has therefore reached "the height of unhappiness," he bemoans, for folks are "not only attracted, but even pleased, by shameful things." Consequently, what once were considered a man's depraved vices have now become his daily habits (*Ep.* 39.6; LCL).[1]

Like Käsemann, Anton Fridrichsen also gestures to Stoic works to show how "the image of the moral judge who does what he blames on others was a widespread *topos* in satire and moral sermons" in the first century (Fridrichsen 1927, p. 41). According to Fridrichsen, within these works, the philosophers often demand that there be a "consistency between theory and practice" as they underline the contrast between "the inner and outer

man." Moreover, Fridrichsen remarks that the Stoics here also tend to warn about a person's weakness for praise and applause as well as her vulnerability to ridicule from others. Rather than the foolish concern for the opinion of man, the Stoics stressed "an individual's awareness of her inner quality and its harmony with nature and God" (Fridrichsen 1927, pp. 45–48).

To instantiate, Fridrichsen refers to how what Seneca writes in *Vit. Beat* 27.4–6 rings conspicuously consonant with Romans 2 (Chaumartin 1985; Hine 2017; Inwood 2005). In that section, the Stoic censures his imaginary opponent: "But as for you," the sage asks, "how do you have the leisure to search out others' evils and to pass judgement upon anyone?" However, rather than the hypocrite minding his own business, according to Seneca, he abuses his betters with bitter questions such as, "Why does this philosopher have such a spacious house?" and "Why does this one dine so sumptuously?" In response, the sage proclaims the following to his interlocutor: "You look at the pimples of others when you yourselves are covered with sores." Not done, Seneca asks the hypocrite why he criticizes others instead of examining his own sins that rend him on every side—those "assailing you from without," or attacking from within—which rampage and rage "in your very vitals." The Stoic goes on to land a final blistering blow:

> Human affairs—even if you have insufficient knowledge of your own position—
> have not yet reached the situation in which you may have such superfluity of
> spare time as to find leisure to wag your tongue in abusing your betters. (*Vit. Beat*
> 27.4–6; LCL)

Numerous scholars have since followed suit with Bultmann, Käsemann, and Fridrichsen in recognizing how, in Romans 1–2, "Paul follows the practice of contemporary popular preachers of Stoicism" (Dodd 1932, p. 30; Swancutt 2004, pp. 42–73; Dodson 2017b; Thorsteinsson 2013, p. 147; Huttunen 2010; Furnish 1986, pp. 52–82; Martens 1994, p. 94; Schreiner 1998, p. 96; Morris 1988, p. 94; Jewett 2007, pp. 221–23). Nonetheless, while the arguments and proposed parallels from these scholars are helpful and convincing, a significant parallel to Paul's polemic against his fictitious opponent in Romans has been largely neglected, namely, Seneca's reference to the hypocrite in *Epistle* 59.

> Yonder person hears himself called "most gentle" when he is inflicting tortures,
> or "most generous" when he is engaged in looting, or "most temperate" when he
> is in the midst of drunkenness and lust.[2] (*Ep.* 59:11)

Additionally, as will be demonstrated below, this parallel to Rom 2:17–24 gains gravity because the larger context of *Epistle* 59 also regards human depravity.

Therefore, in this article, I seek to follow the examples and buttress the conclusions from the scholars mentioned above (as well as some mentioned below) with respect to how well Romans 2 aligns with passages from Seneca's pen. However, while the *prima facie* resonances have led some to conclude that Paul sounds just like Seneca and that his diatribe fits the classic Stoic indictment of the pretentious philosopher (*infra*) (Stowers 1981, p. 112),[3] I will argue the convergence of *Epistle* 59 with Romans 2 ultimately elicits how significantly the apostle's version varies from that of Seneca. As we will see, in contrast to Paul's invective, Seneca actually shows more solidarity with his hypocritical interlocutor by recognizing his own failings and shortcomings. To this end, then, I will (1) first summarize more recent scholarship that considers the similarities of Seneca's writings with Romans 1–2. Then, (2) I will discuss the relevant passages from Seneca's *Epistle* 59 in order to (3) place them in comparison with Paul's comments in Romans 2—noting both resonances and differences (Barclay 2020; Dodson and Briones 2017, pp. 1–21). Once this has been done, (4) I will be in a better position to make final conclusions.

Before proceeding on, it should be understood that, for the sake of brevity, I will avoid rehashing much of the exegesis of Romans 2 detailed in commentaries. Moreover, the aim here is not "parallelomania," (Sandmel 1962)[4] nor is it to make apologetic proclamations with respect to the "uniqueness" of Christianity, nor to argue for the dependence of Seneca on Paul or vice versa. Regarding the later, I prefer to follow what Jonathan Z. Smith refers

to as "analogous processes," which responds to "parallel kinds of religious situations" rather than the constructing of "genealogical relations" between the two works (Smith 1990, pp. 112–13; Frankfurther 2012). Further, to borrow from Hans Joachim Eckstein, this article seeks to avoid the two extremes of claiming that Paul speaks "exactly like a Stoic,"[5] and that of altogether ignoring "the conceptual influence and adoption from popular philosophy" by Paul (Eckstein 1983, p. 150).

## 2. Current Scholars Comparing Seneca's Works with Romans 1–2

In this section, I will point to works by Michael Wolter and Stanley Stowers to exemplify how contemporary scholars have followed Bultmann, Käsemann, and Fridrichsen in finding resonances of Romans 2 in Stoic writings. For instance, Wolter adds Seneca's *De Clementia* to the collection, the essay in which the sage attempts to convince Nero how people who accuse others are rarely themselves free from blame (Wolter 2014, p. 167). In fact, Seneca states, if put on trial themselves, very few judges would escape the same conviction under the very law they cite for the verdict they deliver. What is even more ironic, the Stoic continues, is that no person is more reluctant to pardon others than he who again and again has had reason to seek it for himself. In truth, though, Seneca declares: "We all have sinned, and shall go on doing so for the rest of our lives" (*Clem.* 1.6.3–4).

American, Australian, and British scholars such as N.T. Wright, Brendan Byrne, and Ben Witherington have followed their German counterparts by bringing up Seneca in their discussions of Romans 2 as well (Wright 2013, p. 2:1088; Byrne 1996, p. 72; Witherington 2004, pp. 69–70). Stanley Stowers stands as the leader of this pack in showing how what appears in Romans 2 would have been standard in Stoicism (Stowers 1994, pp. 100–1, 149). In his argument, Stowers refers to Seneca's *Epistle* 77, where the sage rebukes an interlocutor so besotted with luxury that the glutton can think of nothing else. In that essay, the sage writes the following:

> It makes no difference whether a hundred
>
> or a thousand measures pass through your bladder;
>
> you are nothing but a wine-strainer.
>
> You are a connoisseur in the flavour of the oyster, and of the mullet;
>
> your luxury has not left you anything untasted for the years that are to come;
>
> and yet these are the things from which you are torn away unwillingly.
>
> What else is there which you would regret to have taken from you?
>
> Friends? But who can be a friend to you?
>
> Country? What? Do you think enough of your country to be late to dinner? (*Ep.* 77.16–17)

After Stowers discusses how this passage parallels Paul's, he continues to note the similarities with the other missiles Seneca launches at the interlocutor in the epistle:

> The light of the sun? You would extinguish it, if you could;
>
> for what have you ever done that was fit to be seen in the light?
>
> Confess the truth; it is not because you long for the senate-chamber or the forum,
>
> or even for the world of nature, that you would fain put off dying;
>
> it is because you are loth to leave the fish-market, though you have exhausted its stores.
>
> You are afraid of death; but how can you scorn it in the midst of a mushroom supper?
>
> You wish to live; well, do you know how to live?
>
> You are afraid to die. But come now: is this life of yours anything but death? (*Ep.* 77.18)

From this and other examples, Stowers infers that "Paul's characterization of the pretentious person" in Romans 2 is consonant with this popular Stoic polemic used to attack a "hypocritical moralist" so as to expose "the difference between word and deed in people's lives" (Stowers 1981, p. 83). This well-known trope portrays a man who is "above all a boaster and someone who pretends to be what he is not"(Stowers 1994, p. 145). (For example, along these lines, Theophrastus portrays a man who not only claims to own a palatial house that in reality he only rents but also tells others he plans on selling the plush estate because it is too small (Thiessen 2016, p. 62)[6]. Further, according to Stowers, the Stoic tends to depict the character as grossly conceited, arrogant, and proud—one who despite his boasts about "the external trappings of wealth, honor, power, or virtue" fails to possess them (Stowers 1994, p. 145). Stowers deduces that the man portrayed in Rom 2:17–29 belongs to this ubiquitous type (Stowers 1994, p. 145). In fact, he goes so far as to conclude: "If the reference to a Jew [in Romans 2] were changed to a Stoic and the obvious Jewish references in vss. 22b and 24 were eliminated, then this text would be a classic example of indictment of the pretentious philosopher" (Stowers 1981, p. 112).

Considering these works, I propose Seneca's *Epistle* 59 be added as another parallel with Romans 2, which, in turn, supports the conclusions from the aforementioned scholars all the more. To clarify, I will focus on how the epistle provides a contemporaneous comparison that elucidates further evidence that Paul's strategy was to draw upon common Stoic tropes to land his rhetorical blows. In contrast, however, I will attempt to show that the similarities of the two works highlight their discontinuity. To this end, in the next section, I will first discuss the relevant passages from Seneca's *Epistle* 59.

### 3. The Yonder Man in Seneca's *Epistle* 59

Seneca likely wrote his epistles between 62–64 CE, mostly from Rome.[7] Debate remains regarding whether these letters reflect a real correspondence between Seneca and Lucilius or whether the epistolary form represents a literary fiction (Setaioli 2014a). Either way, Senecan scholars tend to agree the Stoic meant for these epistles to be "open letters" available to a wider audience—"including posterity, which Seneca considered his ultimate addressee" (Setaioli 2014a, p. 194). With respect to style, his epistles exhibit not only a "declamatory accent" that prioritizes the brilliant quip over a "steadier consistency," but they also reflect diatribic features such as "colorful imagery, rhetorical questions and exclamation, interlocutory invention and sharp rejoinder, rapid-fire imperatives, and sharp denunciations of vice" (Williams 2015, p. 139).

According to Runar Thorsteinsson, Seneca's epistles stand as the most significant and complex in Greco-Roman letters because of the "more extensive, calculated, and consistent employment of an interlocutor" (Thorsteinsson 2003, p. 134). Accordingly, Thorsteinson provides some helpful statistics. For example, "out of a maximum of 100 letters in which a conversational partner is surely invented [in Seneca's letters], at least 65 include the recipient, Lucilius, exclusively as the interlocutor" (Thorsteinsson 2003, p. 142). Furthermore, Thorsteinson continues: "in approximately 15 letters one interlocutor represents Lucilius while another (or others) represent someone else" (Thorsteinsson 2003, p. 142). Thorsteinsson goes on to warn that sometimes "the exact boundaries" between one or more interlocutors in Seneca's letters can be "very vague" (Thorsteinsson 2003, p. 143).

When Seneca uses the dialogical style, to borrow from Bardo Maria Gauly, his *eigentliches Ziel* is to help the readers rid themselves of the *Schmutz der Welt* (Gauly 2004, p. 32). This aim can be seen in *Ep.* 59:9–10, where the Stoic denounces humankind's widespread, deep-seated *Schmutz.* Seneca writes:

> We human beings are fettered and weakened by many vices;
>
> we have wallowed in them for a long time,
>
> and it is hard for us to be cleansed.
>
> We are not merely defiled [by sins];
>
> we are dyed by them. (*Ep.* 59.9)[8]

This warped condition leads the sage to confess the sober question he frequently considers in his heart, namely, "why is it that folly holds us with such an insistent grip."[9]

Seneca infers that the answer to this riddle lies in the fact that people do not battle against vice strongly enough, because—he figures—"we do not struggle towards salvation with all our might."[10] What is more, he continues, "we do not put sufficient trust in the discoveries of the wise, and we do not drink in their words with open hearts."[11] Therefore, the Stoic sighs, "None of us goes deep below the surface" but only skim from the top, approaching this great moral problem "in too trifling a spirit."[12] In fact, Seneca quips, the only time a man sets aside to learn how to overcome his vices is the little time he has left over from indulging them (59.10) ([Bellincioni 1979](#), pp. 255–56).

For the sage, though, the greatest obstacle to humanity's freedom is that "we are too readily satisfied without ourselves" because we believe the shameless flattery heaped upon us from others. As a result, he explains, when someone "calls us good men, or sensible men, or holy men, we see ourselves in his description"(*Ep.* 59.11). However, how ridiculous we are, Seneca continues, to "agree with those who declare us to be the best and wisest of men," even though we know these people who declare these things are known to be liars (*Ep.* 59.11). What is even more asinine, he reasons, is that "we desire praise for certain actions when we are especially addicted to the very opposite actions."[13] To illustrate this phenomenon, Seneca continues:

> Yonder person hears himself called "most gentle"
>
> when he is inflicting tortures,
>
> or "most generous" when he is engaged in looting,
>
> or "most temperate" when he is in the midst of drunkenness and lust. (*Ep.* 59:11)[14]

Consequently, the sage deduces, "it follows that we are unwilling to be reformed, just because we believe ourselves to be the best of men." Therefore, he contends, people need humble self-awareness to redress the ludicrous self-deception that marks us. The Stoic illustrates what he means with a story about how Alexander the Great, after he was pierced by an arrow and forced to retreat from combat, blurted the following: "All men swear that I am the son of Jupiter, but this wound cries out that I am mortal."[15] Thus also—Seneca writes—in this modest vein, if anyone calls us wise, we should promptly protest

> 'You call me a man of sense,
>
> but I understand how many of the things which I crave are useless,
>
> and how many of the things which I desire will do me harm.
>
> I have not even the knowledge, which satiety teaches to animals,
>
> of what should be the measure of my food or my drink.
>
> I do not yet know how much I can hold'. (*Ep.* 59.13)[16]

In short, rather than be stultified by flattery, a person should be steeped in humility, like Seneca. To assist in this pursuit, the Stoic states to his interlocutor, "I shall now show you how you may know that you are not wise (*Ep.* 59.14). Seneca remarks that, in comparison to the interlocutor's demeanor, "the wise man is joyful, happy, calm, and unshaken (*Ep.* 59.14). He continues:

> Now go, question yourself; if you are never downcast,
>
> if your mind is not harassed by apprehension . . .
>
> if day and night your soul keeps on its even and unswerving course with itself,
>
> then you have attained to the greatest good that mortals can possess. (*Ep.* 59.14)

Moreover, the sage goes on to say to his interlocutor, when you are looking for joy in all the wrong places, you have fallen short of wisdom and strayed from its path (*Ep.* 59.14). To clarify his interlocutor's true position, Seneca informs him as follows:

You must know that you are as far short of wisdom as you are short of joy.

The Stoic then chides him again. He writes, while joy is the goal, "these objects for which you strive so eagerly, as if they would give you happiness and pleasure, are merely causes of grief."[17] In other words, to borrow from F. Scott Fitzgerald, the Stoic disdains the foul dust that floats in the wake of the yonder man's actions, which are nothing more than "the abortive sorrows and short-winded elations of men" (Fitzgerald 2018, p. 2). Seneca goes on to write

All men of this stamp, I maintain, are pressing on in pursuit of joy,

but they do not know where they may obtain a joy that is both great and enduring.

One person seeks it in feasting and self-indulgence;

another, in canvassing for honours and in being surrounded by a throng of clients;

another, in his mistress;

another, in idle display of culture and in literature that has no power to heal;

all these men are led astray by delights which are deceptive and short-lived—

like drunkenness for example, which pays for a single hour of hilarious madness

by a sickness of many days. (*Ep.* 59.15)[18]

The sage then sets these "pleasure lovers" over against the wise man who is "never deprived of joy," since true joy "springs from the knowledge that you possess the virtues." Thus, Seneca reasons, only "the brave, the just, the self-restrained, can rejoice."[19]

Seneca anticipates his interlocutor's rebuttal, though, and gives it voice.

But you ask,

'What do you mean?

Do not the foolish and the wicked also rejoice?'

The Stoic answers the question by calling his reader to be realistic; surely such people "rejoice" in carnal pleasures, which results in misery not joy. Seneca explains,

When men have wearied themselves with wine and lust,

when night fails them before their debauch is done,

when the pleasures which they have heaped upon a body that is to small to hold them begin to fester, at such times they utter in their wretchedness those lines of Vergil:

'Thou knowest how, amid false-glittering joys,

We spent that last of nights [before the fall of Troy]'. (*Ep.* 59.18)

In contradistinction to these wretches, the sage declares, those who imitate the gods gain everlasting delight. So, over against the pursuit of passing pleasures, he concludes, "the joy which comes to the gods, and to those who imitate the gods, is not broken off, nor does it cease."[20]

Now that we have summarized the most relevant passages in *Epistle* 59, we are in a better position to place it in comparison with Romans 2.

### 4. The Hypocrite in Romans 2:17–24

*4.1. Romans 2:1–5*

After tracing out humanity's chronic depravity in Rom 1:18–32, Paul turns on his interlocutor in 2:1–5, who—as Fridrichsen puts it—is "*kein Haar besser*" than the depraved pagans portrayed in the previous passage, since the man judges the wicked while practicing the same sins they do (Fridrichsen 1927, p. 41). Thereby, according to the apostle, this hypocrite condemns himself so that—like the fools in 1:20—he, too, is without excuse (Fridrichsen 1927, p. 41). Although the judge commits sins like the ungodly in chapter 1, he thinks he will escape judgment. Therefore, to borrow from David Garland, his crucial mistake "is not that he dares to mount the judge's throne or to gloat hauntingly over God's

condemnation of others." Rather, the man "fails to recognize that he is also guilty of sins that incite God's wrath and fury"(Garland 2021, p. 89). Furthermore, Paul tells the judge that he shows contempt for the Lord's forbearance and patience when he fails to realize that God's kindness is meant to lead him to personal repentance—not false-assurance or scorn.

While a considerable number of scholars, if not the majority, see 2:1–5 as well as 2:6–24 as falling within "*der innerjüdische Israel-Diskurs*,"(Wischmeyer 2006, p. 359; Linebaugh 2013, p. 101) others strongly reject the notion that Paul targets an ethnic Jewish person here. The latter argue instead for a gentile "Judaizer" who calls himself a Jew (Fredriksen 2017, p. 157), or better put—to draw from Rafael Rodríguez—one who is "a Jew religiously but is a gentile ethnically" (Rodríguez 2014, p. 51). It is beyond the scope of this article to decide which of these is the best option (Oropeza 2021; Watson 2007, pp. 200–201, n. 20). Nor is it crucial to say whether or not 2:17–24 continues the diatribe Paul started in 2:1–5.[21] Whichever of these interpretations one lands on, though, an anti-Semitic reading must be rejected, since, as Friedrich Wilhelm Horn puts it, Paul's "polemics are of course not aimed at every Jew or even at Judaism in general." Rather, as Horn continues, we must remember the apostle—himself a "Hebrew of Hebrews"—is attacking a particular *Gesprächspartner*, "who because of his special position with the Torah," claims superiority over everybody else (Horn 2011, p. 218).

### *4.2. Romans 2:6–11*

According to Leander Keck, Paul goes on in vv. 6–11 to stress his points to the interlocutor by using a chiastic structure (Keck 2005, p. 77).

A    ὃς ἀποδώσει ἑκάστῳ κατὰ τὰ ἔργα αὐτοῦ·(v. 6)
B    τοῖς . . . ἀφθαρσίαν ζητοῦσιν ζωὴν αἰώνιον (v. 7)
C    τοῖς . . . ἀπειθοῦσι τῇ ἀληθείᾳ πειθομένοις . . . ὀργὴ καὶ θυμός (v. 8)
C′    θλῖψις καὶ στενοχωρία ἐπὶ πᾶσαν ψυχὴν . . . τοῦ κατεργαζομένου τὸ κακόν (v. 9)
B′    δόξα δὲ καὶ τιμὴ καὶ εἰρήνη παντὶ τῷ ἐργαζομένῳ τὸ ἀγαθόν . . . (v. 10)
A′    οὐ . . . ἐστιν προσωπολημψία παρὰ τῷ θεῷ (v. 11)

So then, with this rhetorical device, the apostle underscores that the Lord will repay each person according to her deeds, for God does not show favoritism (A, A′). Further, while divine judgement results in glory and immortality for those doing good (B, B′), it brings wrath, anguish, and fury for those who walk in disobedience (C, C′) (Collins 2010).

### *4.3. Romans 2:12–16*

Verses 12–16 comprise the third section of Paul's argument he began in 2:1 (Longenecker 2016, p. 271). Here the apostle explains the meaning of his aphorism in v. 11, οὐ γάρ ἐστιν προσωπολημψία παρὰ τῷ θεῷ. To do so, Paul introduces the Torah for the first time in the letter so as to distinguish between those without the Law (ἀνόμως) and those in the Law (ἐν νόμῳ). Both groups, the apostle declares, will be held accountable for their disobedience or be justified by their obedience. The apostle declares that the Lord will even justify non-Jews who, despite their disadvantage of not having the Law by nature (φύσει), still do what the Torah commands and thereby demonstrate τὸ ἔργον τοῦ νόμου γραπτὸν ἐν ταῖς καρδίαις αὐτῶν. Paul concludes this subsection with a reference to the final day when God will show no partiality in judging the secrets of men—a verdict rendered κατὰ τὸ εὐαγγέλιόν μου διὰ Χριστοῦ Ἰησοῦ (2:16).

### *4.4. Romans 2:17–24*

This brings us to our key passage, vv. 17–24. Michael Gorman divides the section into "three groupings of five,"—five phrases to describe the interlocutor's identity, five phrases to define his mission, and five queries to expose his hypocrisy and pride (Gorman 2022, p. 101). First, the apostle identifies the man as someone who calls himself a Jew, who trusts in the Law, who boasts in God, who knows the divine will, and who discerns and approves what is excellent so as to leave the *adiaphoria* behind (Dunn 1988, p. 111; Moo 2018, pp. 170–71). As J.M. Diaz-Rodelas puts it, now that Paul has touched on "self-awareness" (*la*

*autoconciencia*) of the one calling himself a Jew, in vv. 19–20, he shifts to focus on the man's mission to "the non-Jewish world" (*la relación de la misma con el mundo de los no judíos*) (Diaz-Rodela 1994, p. 84).

According to Paul, with respect to this mission, the interlocutor remains convinced that he is a guide to the blind, a light in darkness, an instructor of the ignorant, and a teacher of the immature. What is more, he shamelessly considers himself to have the embodiment of knowledge and truth. Finally, the apostle rattles off five rhetorical questions to lay his opponent bare:

ὁ οὖν διδάσκων ἕτερον σεαυτὸν οὐ διδάσκεις;

ὁ κηρύσσων μὴ κλέπτειν κλέπτεις;

ὁ λέγων μὴ μοιχεύειν μοιχεύεις;

ὁ βδελυσσόμενος τὰ εἴδωλα ἱεροσυλεῖς;

ὃς ἐν νόμῳ καυχᾶσαι, διὰ τῆς παραβάσεως τοῦ νόμου τὸν θεὸν ἀτιμάζεις; (SBL)[22]

You, then, that teach others, will you not teach yourself?

While you preach against stealing, do you steal?

You that forbid adultery, do you commit adultery?

You that abhor idols, do you rob temples?

You that boast in the law, do you dishonor God by breaking the law? (2:21–23; NRSV)

The first of these questions stands as the heading for the following ones which stress how the hypocrite fails to teach himself (Moo 2018, p. 173). Horn as well as Matthew Theissen demonstrate how the three vices of theft, adultery, and sacrilege (Kruse 2012, pp. 150–51)[23] comprise a common trope in the "*polemisches Katalogmaterial*" of moral philosophy, going at least as far back as Aristotle and used to exemplify a person's preaching one thing but doing another.[24] (The same three vices also occur together in Seneca's *Ep.* 87.23).

To cap off the attack, the apostle then claims his interlocutor has indeed fulfilled Scripture but not in the way the hypocrite had presumed. Instead of his exalting the Lord, Paul's interlocutor has caused the nations to blaspheme God's name: τὸ γὰρ ὄνομα τοῦ θεοῦ δι' ὑμᾶς βλασφημεῖται ἐν τοῖς ἔθνεσιν, καθὼς γέγραπται (v. 24). Keck avers that, with this quotation, Paul places the hypocrite "in essentially the same position" as the idolators in 1:21 who failed to honor God as God (Keck 2005, p. 83).

### 4.5. Romans 2:25–3:9

Although this article focuses on vv. 17–24, it will be helpful to summarize Paul's argument following this passage. In 2:25–29, similar to 2:12–16, Paul places the uncircumcised who obey the law in contradistinction to the circumcised who fail to obey it. The former will judge the latter who break the covenant requirements despite their circumcision. For Paul, what matters entails not what a person is outwardly, according to the flesh. Instead, the apostle promotes ὁ ἐν τῷ κρυπτῷ Ἰουδαῖος, whose heart has been circumcised by the Spirit and whose praise comes from God not people (οὗ ὁ ἔπαινος οὐκ ἐξ ἀνθρώπων ἀλλ' ἐκ τοῦ θεοῦ).

In 3:1–9, Paul gives voice to his opponent's objections: is there an advantage in being a Jew and in circumcision; does the unfaithfulness of the unfaithful nullify the faithfulness of God; is God unjust in his judgements. The apostle answers the three diatribal questions with (1) "yes," there is an advantage to being a circumcised Jew, (2) "no", people's unfaithfulness do not nullify God's faithfulness, and (3) "yes indeed," God remains just in his judgment of the world. With a catena of texts from both the Psalms and Isaiah, the apostle proceeds to prove in a *Schlussplädoyer* that both the gentiles and the Jews are ὑφ' ἁμαρτίαν (Horn 2011, p. 218). As a result, no one is righteous or good. No one understands or seeks God.

But rather, with mouths full of curses, they all practice violence and deceit. They do not fear God or know peace. Instead ruin and misery mark their ways (3:10–18).

Having now looked at both Seneca's *Epistle* 59 and Romans 2, we are ready to compare them.

### 5. *Epistle* 59 and Romans 2

In this section, I will place Seneca and Paul's thoughts on (1) hypocrisy and (2) human depravity side-by-side in order to show that there is enough overlap to support *Epistle* 59 as a viable parallel to Romans 2—enough, at least, to add it to the list of other Senecan works used to suggest, as stated above, that Paul drew upon common Stoic tropes to land his rhetorical blows.

### 5.1. Hypocrisy

First, Seneca's comments on hypocrisy in *Epistle* 59 fit the popular *topos* that Fridrichsen and Stowers detail regarding the Stoic's depiction of a person who pretends to be someone he is not. For Seneca, this character is the yonder man who accepts the reputation of being "most gentle," despite the reality that he inflicts torture on others. Moreover, this hypocrite also embraces the praise that he is "most generous," while he is simultaneously engaged in looting. Additionally, the yonder man believes others who call him "most temperate" even while he lives in drunkenness and lust.

Likewise, as just seen above, the apostle calls out the man who boasts that he knows what is best. This man further brags about how he relies on the Law so that he serves as a guide to the blind, a light to those in darkness, a teacher of children and corrector of fools. Paul then discharges a series of rhetorical questions to upbraid his interlocutor for not practicing what he preaches so that he himself steals, fornicates, robs temples, and breaks the Law. In comparison, whereas both the apostle and the sage uncover the hypocrisy of their respective targets, there is a stark difference. That is, Seneca's yonder man fails to live up to the reputation given to him by others, but Paul's interlocutor brags on himself, calling himself a Jew (Εἰ δὲ σὺ Ἰουδαῖος ἐπονομάζῃ), particularly one who embodies knowledge and truth.[25] In this sense, the apostle's boaster is worse off than Seneca's yonder man, in that the interlocutor in Romans not only steals, commits adultery, and robs temples, but he does so while he *preaches* against these very things. In other words, the sage's yonder man is *not* said to teach against torture, looting, and lust: his fault is that he believes the flattery that he is a moral man despite his pursuits of vice. In juxtaposition, then, Seneca's caricature suffers more from ironic ignorance or deliberate self-delusion than Paul's man who, full of egregious hypocrisy, with the Torah in his hand, commits the very sins he publicly denounced.

### 5.2. Human Depravity

The context of the passage surrounding Seneca's yonder man makes the proposed parallel with Romans even more valid. As demonstrated above, according to the sage, humanity is not only chained to and hampered by vices, but also defiled and dyed by them. It is therefore difficult, Seneca reasons, for people to be cleansed, especially since no one exerts great effort to be free, nor trusts the words of the wise, nor drinks their truth to the dregs. Instead of passionately pursuing salvation with all their might, mankind seeks joy in feasting and self-indulgence, in honors and popularity, in mistresses and so on. Consequently, rather than imitating the gods which leads to unceasing and unbroken joy, people are led astray by deceptive and evanescent delights, which—as Seneca so markedly puts it—"is like drunkenness in that it pays for a single hour of hilarious madness by a sickness of many days" (*Ep.* 59.15) (Brennan 2009). The Stoic also gripes that nobody strives for salvation with all their might, nor even has the good sense of a savage beast who knows its own limitations (*Ep.* 59.9, 13). According to the sage, people are worse than the wild animals, since their overindulgence results in their craving not only what is useless, but what is harmful, too (*Ep.* 59.17). What humanity supposes will provide them happiness,

Seneca insists, are worse than passing pleasures. In truth, they are the consequential causes of the world's persistent griefs (*Ep.* 59.14). Discussions of depravity also surround Paul's diatribe in Romans 2. While the sage concludes that people are profoundly corrupted by vice and remains in its custody, the apostle goes on to declare that all people were under the power of sin (πάντας ὑφ' ἁμαρτίαν; 3:9–10). The unredeemed masses are so worthless (ἠχρεώθησαν) that no one seeks God (οὐκ ἔστιν ὁ ἐκζητῶν τὸν θεόν), that no one is righteous (Οὐκ ἔστιν δίκαιος), and that no one has understanding (οὐκ ἔστιν ὁ συνίων).

Paul proclaims that God will repay each person according to his or her deeds. Those who seek glory, honor, and immortality stand over against those who are wicked, self-seeking (ἐριθείας), and disobedient to the truth. Whereas the former will receive eternal life, for the latter there will be wrath, fury, anguish, and distress (2:7–10). Since Seneca's conception of the afterlife differs from Paul's and because the Stoic did not believe in a God who gets angry or pours out wrath, the fate of Seneca's fools is that they simply flit around false-glittering pleasure and never find lasting joy. This consequence seems mild compared to Paul's warning that pleasure seekers will not only fail to find lasting life but receive divine condemnation instead. Along these lines, while the apostle frets that humans do not seek God, for the philosopher, the goal is not the gods themselves but the joy that comes from imitating them (Rowe 2016, pp. 1–258).[26] Whereas Seneca cites Vergil to liken the fools with those who partied on the night before the fall of Troy, Paul's hypocrite has not only fallen short of God's glory—along with unredeemed humanity—but even worse, he represents those ill-fated people whom the prophet Isaiah proclaimed had also sullied God's honor and reputation (τὸ γὰρ ὄνομα τοῦ θεοῦ δι' ὑμᾶς βλασφημεῖται ἐν τοῖς ἔθνεσιν, 2:24).

Drawing from Mikhail Bahktin, the juxtaposition above leads to the intersection where the texts of Seneca and Paul come to life as they collide and flash to illuminate both Stoic and Pauline thought more brightly (Baktin 1986, p. 162). This light shows how Seneca identifies more with the plight of the hypocrite in vice's possession, while Paul identifies with the righteous who have been set free from sin's rule. That is to say, Seneca wonders why vice holds "us" in its grip. His frustration is that "we" do not pursue salvation with all "our" strength; that "we" do not put sufficient trust in the words of the wise; that "we" approach the predicament in too trifling of a spirit; that "we" long for people to praise "us" for certain actions while "we" do the very opposite; "we" are not sensible because "we" crave that which is worthless and that does "us" harm.

In this case, Seneca sounds more like Paul's wretch in Romans 7 complaining of moral impotence than Paul's remarks about the righteous in Romans 2—who, with circumcised hearts, keep the requirements of the Law and receive praise from God rather than from people (2:26–29) (Wolter 2014, p. 168). In contradistinction to Seneca, then, instead of those under sin, the apostle associates with those who have been justified and redeemed by grace (3:24), renewed and set free from Sin's grasp (6:1–14), who are no longer obliged to fulfill their ungodly desires (8:12), and who make no provision for the flesh since they have put off vice and have put on Christ (13:13–14).

While Paul likely does not have Seneca in mind as the interlocutor (as some scholars propose, which violates the analogous process),[27] the apostle's bolt aimed at his interlocutor would likely hit close to Seneca's home, since elsewhere the Stoic is forced to respond to accusations from his opponents for preaching one thing while doing another (*Vit. Beat.* 18.1). Strikingly similar to the comments about his hypocrite in *Epistle* 59, Seneca's foes point out how he is really a coward despite his façade; that he pursues the applause of man while claiming not to care about his reputation; and that he lavishes himself in luxury despite his appeals for people to live simply.[28] Moreover, Seneca's replies in *De vita beata* resonate with the "we" and "us" passages in *Epistle* 59. In the former, he admits that he has never attained wisdom, that he continues to be inflicted with a moral abscess, and that he remains fastened to his cross despite his attempts to be liberated from its sinful beams (*Vit. Beat* 18.1–19.3).[29]

To be sure, these statements in *Epistle* 59 and *De vita beata* accord with the tenor of Seneca's works in general. As Aldo Setaioli underlines, "It must in fact be stressed from the outset that Seneca considers himself to be in need of self-improvement no less than the people he addresses in his writings" (Setaioli 2014b, p. 239; Schofield 2009, pp. 233–56; Holladay et al. 2014, p. 711). The Stoic presents himself to his reader as a fellow traveler on "the same therapeutic path of self-improvement," Williams (2015, p. 135) who—to borrow from Anne-Marie Guillemin—as a *directeur d'âmes* wanted his ethical pleas to be pursued, even if he himself had not yet achieved them (Guillemin 1952). For his audience, then, he is both the "*Instrument philosphischer Erziehung und Selbsterziehung*" (Myers 2022, pp. 1–196). Conversely, while Paul too serves as a *directeur d'âmes*, he guides his audience with a steady confidence that the law of the Spirit of life in Christ Jesus had set him free from the power of Sin (Rom 8:1–4). The apostle therefore directs his churches with the conviction that the righteous requirement of the Law has already been filled in him and in his followers who, putting off the old person, unite with Christ and walk by the Spirit. Over against Seneca and his yonder man, then, Paul's followers have the power to overcome their fleshly passions.

## 6. Conclusions

This article proposes that Seneca's *Epistle* 59 be added to the repertoire of Stoic parallels with Paul's letter to the Romans since, like Romans 2, it lambasts a person whose life does not match his reputation. For example, the yonder man harms others while being called "most gentle"; he robs others even though he is supposed to be "most generous"; and he engages in drunkenness and lust, even though he believes the flattery that he is "most-temperate."

Beyond surface similarities, there are of course also distinct differences between the two. Among other dissimilarities mentioned above, perhaps the most indelible is Seneca's identification with the hypocrites. Like them, he too has *not* been liberated from folly and vice. So, what most sets Seneca apart from his interlocutor is that he realizes his hypocrisy and approaches this moral inconsistency with honesty and humility. As demonstrated, this is where the convergence between his and Paul's two works highlights their great divergence. In comparison to Seneca and his hypocrites, the apostle excoriates his boasting interlocutor from a position of confidence, integrity, and moral freedom. Sin is no longer the boss of Paul or of his fellow believers. Having put away the old life that was enslaved to the flesh, they have been clothed in Christ so that they can walk in the obedience that comes from faith (Von Albrecht 2004, p. 2). The intellectual rigor through which the Roman Stoic seeks freedom stands in contradistinction of the Christian grace that liberates the Roman church.[30] What is more, this liberty so rarely (if ever) realized by the Stoics remained ready at hand for the believers.[31]

In conclusion, then, Paul's assault on the moralist who lives in the opposite manner of his reputation coheres with that of Seneca's. In contrast to the apostle, however, when Seneca rages against the latter, he sees himself in them. So then, while the comments in *Epistle* 59 support the arguments of the aforementioned scholars regarding Stoic resonances in Romans 1–2, the parallel reminds us that even if Paul draws upon Stoic ideas and rhetorical devices to deride the so-called Jew, the apostle considers his followers as standing above the Stoic philosophers too. In other words, Seneca's yonder man is comparatively not that far yonder from the Stoic's own failings, but Paul's interlocutor could not be much farther from those Jewish believers and righteous Gentiles who have been united with Christ. Therefore, the apostle would consider himself and his Roman congregation as not only distant from the likes of the pretentious yonder man, but from the humble hypocrisy of Seneca, too.

**Funding:** This article received no external funding.

**Institutional Review Board Statement:** Not applicable.

**Informed Consent Statement:** Not applicable.

**Data Availability Statement:** Not applicable.

**Acknowledgments:** I would like to thank my graduate assistant, Samuel Woo, for his valuable help with this essay.

**Conflicts of Interest:** The author declares no conflict of interest.

## Notes

1. *Epistle* 39.6, emphasis mine. Translations are from the Loeb Classical Library edition.
2. *Mitissimum ille se in ipsis suppliciis audit, in rapinis liberalissimum, in ebrietatibus ac libidinibus temperantissimum.*
3. Although this essay focuses on Stoic parallels, this is not to preclude resonances in Second Temple literature such as Pss.Sol. 8:8–13; T.Levi 14; CD 4:12–17; 8:4–10. I am indebted to one of my anonymous reviewer's for this qualification.
4. For more on this, see Van der Horst (1980, p. 4); Aune (2014); Malherbe (2013, pp. 679–87). Scholars should also reject parallelophobia; see Huttunen (2009, p. 18).
5. This is Eckstein's citation of Dodd (1932, p. 36).
6. Theophrastus, *Characters* 23.1.
7. For dating, occasion, and genre, see Marshall (2014, pp. 33–44).
8. *Nos multa alligant, multa debilitant. Diu in istis vitiis iacuimus, elui difficile est. Non enim inquinati sumus, sed infecti.*
9. *Quid ita nos stultitia tam pertinaciter teneat?*
10. *Nec toto ad salutem impetu nitimur.*
11. *Quae a sapientibus viris reperta sunt, non satis credimus nec apertis pectoribus haurimus leviterque tam magnae rei insistimus.*
12. *Ep.* 59.10.
13. *Ep.* 59.11; *Adeoque indulgemus nobis, ut laudari velimus in id, cui contraria cum maxime facimus.*
14. See note 2 above.
15. *Ep.* 59.12–13.
16. *Vos quidem dicitis me prudentem esse, ego autem video, quam multa inutilia concupiscam, nocitura optem. Ne hoc quidem intellego, quod animalibus satietas monstrat, quis cibo debeat esse, quis potioni modus. Quantum capiam adhuc nescio.*
17. *Ep.* 59.14; *Ista, quae sic petis tamquam datura laetitiam ac voluptatem, causae dolorum sunt.*
18. *Omnes, inquam, illi tendunt ad gaudium, sed unde stabile magnumque consequantur, ignorant. Ille ex conviviis et luxuria, ille ex ambitione et circumfusa clientium turba, ille ex amica, alius ex studiorum liberalium vana ostentatione et nihil sanantibus litteris; omnes istos oblectamenta fallacia et brevia decipiunt, sicut ebrietas, quae unius horae hilarem insaniam longi temporis taedio pensat.*
19. *Ep.* 59.17.
20. *Quod deos deorumque aemulos sequitur, non interrumpitur, non desinit*; *Ep.* 59.18. Cf. Lee (2020, p. 331); Brookins (2017, p. 185).
21. Debate surrounds this association: e.g., Elliot (1990, p. 127); Rodríguez (2014, pp. 47–71). But others conclude Paul has the same Interlocutor throughout Romans 2: e.g., Wilkins (1997, pp. 121–47); Dunn (1988, pp. 78–82); Gathercole (2002, pp. 198–215); Moo (2018, pp. 136–75); Thorsteinsson (2003, pp. 159–64); Fredriksen (2017, p. 157); Campbell (2009, p. 345); Stuhlmacher (1994, p. 49).
22. This last line could be a statement rather than a question. Cf. UBS and NA28.
23. Regarding how scholars understand "rob temples," see Kruse (2012, pp. 150–51). Cf. Derrett (1994).
24. Thiessen (2016, pp. 60, 63); Horn (2011, pp. 222–25). Cf. Dodd (1932, p. 39); Garland (2021, p. 108); Watson (2007, pp. 203–5); Campbell (2009, p. 561); and Thiessen (2014).
25. This is in taking ἐπονομάζη as a middle. See Longenecker (2016, p. 299).
26. On how Paul and Seneca represent two traditions in juxtaposition, see Rowe (2016, pp. 1–258).
27. Contra Bruce (1985, pp. 93–94).
28. See *Ep.* 63.14. Cf. Griffin (2008); Momigliano (1950); Thorsteinsson (2010); Grimal (1969).
29. Cf. *Ep.* 17.3–4; 68.8.
30. I am indebted to my anonymous reviewer for this insight.
31. For more on Seneca's conception regarding the rarity of the Stoic sage, see Dodson (2017a, pp. 247–66).

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
