# Peer review of "The Yonder Man and the Hypocrite in Seneca’s Epistle 59 and Paul’s Letter to the Romans"

_religions, doi:10.3390/rel14020235_

Round 1
Reviewer 1 Report
This article builds upon a large body of research that has argued that Paul’s Letter to the Romans can be usefully interpreted alongside Stoic motifs and rhetoric. As the author rightly notes, Paul’s language in chapter 2 can be especially fruitfully compared to language found in Stoic authors. This article draws attention to a parallel between a Stoic author and Romans 2 that is often overlooked, namely, Seneca’s Epistle 59. After summarizing recent scholarship that considers similarities between Paul and Seneca, the author outlines Seneca’s letter and draws attention to particular sections that will be of interest to Pauline scholars. In the course of this analysis, the author gives due weight to the way in which Seneca expresses his thought without rushing to compare the letter to Paul. A similar program is followed in the presentation of Romans 2. The author summarizes Paul’s argument in this portion of the letter and allows Paul to speak for himself. After both texts have been outlined, the author compares Paul and Seneca regarding their treatments of hypocrisy and human depravity in these texts. The conclusion rightly notes the resonance between these texts, but it also gives attention to the ways in which each author expresses himself differently. Seneca positions himself closer to the yonder man than Paul locates himself with regard to his interlocutor, while Paul presents sin as something that is no longer characteristic of himself or his community.
The article clearly positions itself in relation to previous research and concisely states the lacuna that it seeks to fill. Adding Seneca’s Epistle 59 to the list of Stoic sources with which Paul’s language resonates is an important contribution to scholarship on the Pauline corpus, while the conclusion recognizes both ways in which the ancient authors converge (their topic and manner of speech) and diverge (the position of the speaker and audience vis-à-vis sin) from one another. The article is thoroughly researched, and the summaries of Seneca and Paul provide both a helpful orientation to readers who do not specialize in Pauline studies and an opportunity for scholars who specialize in these epistolary corpora to engage with the author’s argument. The study is conducted with a narrow scope, but it is carried out well and with careful attention to rhetorical details in the respective letters of Paul and Seneca.
Two minor spelling issues might be addressed before the article is finally submitted for publication.
First, in line 435, “The apostles answers” should be corrected to “The apostle answers”.
Second, I note that Virgil is spelled with an i (Virgil; line 513) and with an e (Vergil, line 303). Would it be possible to bring these uses into alignment, or is it the use of the English translation of Seneca’s letters that creates this discrepancy?
Author Response
Thank you so very much for your close examination of my work. I also appreciate you catching my typo and the discrepancy of the Vergil/Virgil spellings (see below). I have redressed both of these in my revision. Thanks again!
First, in line 435, “The apostles answers” should be corrected to “The apostle answers”. Second, I note that Virgil is spelled with an i (Virgil; line 513) and with an e (Vergil, line 303). Would it be possible to bring these uses into alignment, or is it the use of the English translation of Seneca’s letters that creates this discrepancy?
Reviewer 2 Report
I don't have any substantive suggestions. It's a very well-researched and well-defended article, and I'm not aware of anyone who has made the comparison between this text in Romans and Seneca's Epistle 59. I did wonder if the author would consider this the only significant parallel, or if there might be resonances as well with Second Temple Jewish literature such as Pss.Sol. 8:8-13; T.Levi 14; CD 4:12-17; 8:4-10.
Author Response
Thank you so very much for your close examination of my essay. I really appreciate your comment--“I did wonder if the author would consider this the only significant parallel, or if there might be resonances as well with Second Temple Jewish literature such as Pss.Sol. 8:8-13; T.Levi 14; CD 4:12-17; 8:4-10.”
To answer your question: I do *not* think Seneca’s Ep. 59 is the only significant parallel. My PhD thesis actually focused on the parallels in Romans with the Wisdom of Solomon. Nevertheless, this essay merely focuses on the Roman Stoic parallels. I did add a note on page 2 fn.10 (line 87) to clarify for any other reader who might have the same question.
Thanks again!
Reviewer 3 Report
Only a few comments for possible amendment

Author Response
Thank you so very much for you close examination of my work and for the really helpful comments you made. Here are my responses to them.
- I have corrected the typos, spelling mistakes, and awkward wordings you underlined for me.
- I really appreciate your comment regarding “analogous processes” on page 3 (lines 97-104) As a big fan of J.Z. Smith, I am always happy to work his work into my essays. In fact, the footnote in the submitted essay had already cited Drudgery Divine at this very point. To make Smith’s influence more conspicuous though, I added this line.
- Regarding the later, I prefer to follow what Jonathan Z. Smith refers to as “analogous processes,” which responds to “parallel kinds of religious situations” rather than the constructing of “genealogical relations” between the two works. (lines 102-104)
- Although I really like where your comment is leading, due to space, I was not able to go any further on the number of lexical references for “hypocrite,” especially since neither Paul nor Seneca use the word in Romans 2 or Ep. 59.
- In response to your fine suggestion regarding how Paul “viewed himself as standing within Judaism,” I included this highlighted phrase:
- Rather, as Horn continues, we must remember the apostle – himself a “Hebrew of Hebrews” – is attacking a particular Gesprächspartner, “who because of his special position with the Torah,” claims superiority over everybody else.
- A really big thanks for pointing out how it is anachronistic to bifurcate philosophy/theology. I fully agree. This was an embarrassing accident on my part. I think I was trying to be eloquent rather than clear. I redressed the mistake by changing the wording, (which also remedies the other related comments you made about it):
- Drawing from Mikhail Bahktin, the juxtaposition above leads to the intersection where the texts of Seneca and Paul come to life as they collide and flash to illuminate both Stoic and Pauline thought more brightly. (lines 528-530)
- And back to another helpful reminder for the reader regarding “analogous process,” on page 11, line 547, I added:
- While Paul likely does not have Seneca in mind as the interlocutor (as some scholars propose, which violates the analogous process), the apostle’s bolt aimed at his interlocutor would likely hit close to Seneca’s home, since elsewhere the Stoic is forced to respond to accusations from his opponents for preaching one thing while doing another (Vit. Beat. 18.1).
- Also, on your final, very helpful recommendation, I added the following line on p. 13 lines 591-593:
- The intellectual rigor through which the Roman Stoic seeks freedom stands in contradistinction the Christian grace that liberates the Roman church. What is more, this liberty so rarely (if ever) realized by the Stoics remained ready at hand for the believers.
- I too included a footnote here to another work which gets into Seneca’s notion regarding the rarity of the Stoic sage.
Thank you again for working with me. I do believe the essay is better for it.